# The Effect of Probiotics on Health Outcomes in the Elderly: A Systematic Review of Randomized, Placebo-Controlled Studies

**DOI:** 10.3390/microorganisms9061344

**Published:** 2021-06-21

**Authors:** Ashley N. Hutchinson, Cecilia Bergh, Kirsten Kruger, Martina Sűsserová, Jessica Allen, Sophie Améen, Lina Tingö

**Affiliations:** 1Nutrition-Gut-Brain Interactions Research Centre, School of Medical Sciences, Örebro University, 70362 Örebro, Sweden; kirstenkruger8@gmail.com (K.K.); msc.susserova@gmail.com (M.S.); allen.jessicalynn@gmail.com (J.A.); sophieameen1@gmail.com (S.A.); Lina.Tingo@oru.se (L.T.); 2Clinical Epidemiology and Biostatistics, School of Medical Sciences, Örebro University, 70362 Örebro, Sweden; cecilia.bergh@regionorebrolan.se; 3Department of Biomedical and Clinical Sciences, Division of Inflammation and Infection, Linköping University, 50183 Linköping, Sweden; 4Food and Health Programme, Örebro University, 70362 Örebro, Sweden

**Keywords:** probiotics, elderly, gut microbiota, immune function, gut–brain axis, healthy aging

## Abstract

Increasing evidence suggests that probiotic supplementation may be efficacious in counteracting age-related shifts in gut microbiota composition and diversity, thereby impacting health outcomes and promoting healthy aging. However, randomized controlled trials (RCTs) with probiotics in healthy older adults have utilized a wide variety of strains and focused on several different outcomes with conflicting results. Therefore, a systematic review was conducted to determine which outcomes have been investigated in randomized controlled trials with probiotic supplementation in healthy older adults and what has been the effect of these interventions. For inclusion, studies reporting on randomized controlled trials with probiotic and synbiotic supplements in healthy older adults (defined as minimum age of 60 years) were considered. Studies reporting clinical trials in specific patient groups or unhealthy participants were excluded. In addition to assessment of eligibility and data extraction, each study was examined for risk of bias and quality assessment was performed by two independent reviewers. Due to the heterogeneity of outcomes, strains, study design, duration, and methodology, we did not perform any meta-analyses and instead provided a narrative overview of the outcomes examined. Of 1997 potentially eligible publications, 17 studies were included in this review. The risk of bias was low, although several studies failed to adequately describe random sequence generation, allocation concealment, and blinding. The overall study quality was high; however, many studies did not include sample calculations, and the majority of studies had a small sample size. The main outcomes examined in the trials included microbiota composition, immune-related measurements, digestive health, general well-being, cognitive function, and lipid and other biomarkers. The most commonly assessed outcome with the most consistent effect was microbiota composition; all but one study with this outcome showed significant effects on gut microbiota composition in healthy older adults. Overall, probiotic supplementation had modest effects on markers of humoral immunity, immune cell population levels and activity, as well as the incidence and duration of the common cold and other infections with some conflicting results. Digestive health, general-well-being, cognitive function, and lipid and other biomarkers were investigated in a very small number of studies; therefore, the impact on these outcomes remains inconclusive. Probiotics appear to be efficacious in modifying gut microbiota composition in healthy older adults and have moderate effects on immune function. However, the effect of probiotic supplementation on other health outcomes remains inconclusive, highlighting the need for more well-designed, sufficiently-powered studies to investigate if and the mechanisms by which probiotics impact healthy aging.

## 1. Introduction

With improved technology and advancements in modern medicine, life expectancy continues to increase. Although this is a great achievement, the increased life expectancy presents several challenges for society. As the elderly population in developed countries grows, it is estimated that this age group will consume 75% of health care resources by 2030 [1]. Therefore, identifying factors to improve health, quality of life, and independence for the elderly is essential to reduce healthcare costs and societal burden, as well as to promote individual well-being. The focus should not only be on prolonging life, but also on improving its quality. As people age, they become more susceptible to infections and a wide variety of diseases including cardiovascular disease, cancer, and neurodegenerative diseases [2,3,4,5]. However, this increased susceptibility could be reduced by adequately addressing specific risk factors [6]. Aging is associated with a number of physiological changes including metabolic dysregulation, cognitive decline, vascular alterations, and changes in hormone production [7,8,9,10]. In addition, aging is accompanied by complex changes and dysfunction of the immune system, leading to an increase in the concentration of inflammatory markers in the blood, a phenomenon known as “inflammaging” [11,12]. 

Investigation of the factors underlying health and disease in the elderly has also revealed a role for the gut microbiota, a vast array of microorganisms that colonize the gut [13,14,15,16]. The majority of these microorganisms are bacteria (but also viruses, fungi, and protozoa) that have many beneficial interactions with each other and their host. Microbiota are known to impact numerous processes within the gut including the mucosal immune system, digestion, and vitamin synthesis [17,18,19]. The effect of the gut microbiota extends far beyond the gut, as it is implicated in immune-related disorders such as irritable bowel syndrome, diabetes, and low-grade inflammation [12,20,21]. Furthermore, a growing number of studies suggest that the gut–brain axis (GBA), a bidirectional route of communication between the gastrointestinal tract and the central nervous system (CNS) plays a significant role in host physiology and disease, as alterations in gut microbiota composition have been associated with depression, brain development, cognition and neurodegenerative diseases [22,23,24]. 

Several studies have demonstrated that aging is associated with a shift in microbial composition and decreased diversity [25,26]. Characterization of the fecal microbiota composition of 161 elderly from the ELDERMET project revealed that the microbiota composition of elderly is rather distinct compared to younger adults [13]. In the elderly, a lower proportion of the phylum *Firmicutes* was detected compared to younger adults, as well as a greater proportion of *Bacteriodes* spp., supporting previous findings [27,28]. The study also revealed that there is great inter-individual variation in microbiota diversity than what was observed in younger adults. In addition, O’Toole and colleagues observed the increased variation in the microbiota profiles of older adults as well as increases in *Bacteroidetes* in frail elderly compared to younger adults [14]. The microbiota most heavily impacted by aging tend to be the diversity-associated taxa, including *Prevotella* and associated genera. Interestingly, elderly participants also exhibited a reduction in the proportion of *Clostridium* cluster XIVa, including *Roseburia* and *Ruminococcus*, bacteria that are known to produce the short-chain fatty acid (SCFA) butyrate [28]. 

In addition to demonstrating that aging is associated with alterations in the microbiome, a growing number of studies suggest that correlations exist between microbiota composition and aging-associated clinical conditions and diseases. Distinct microbiota profiles have been correlated with *Clostridium difficile* colitis, colon cancer, cardiovascular disease, frailty, and systemic inflammation in the elderly [29,30,31,32,33]. Recently, a role for the gut microbiota has been implicated in neurodegenerative diseases such as Parkinson’s (PD) and Alzheimer’s (AD) disease [34,35,36,37]. Currently, it is difficult to determine if the observed shifts in microbiota composition in age-associated diseases are part of the cause or merely an effect of these conditions and the accompanying lifestyle changes (increased medication, reduced mobility, changes in living arrangement, diet alterations, etc.). However, taken together, these findings suggest that manipulation of the microbiota composition in the elderly is a promising potential strategy to prevent or reduce age-associated disease and disability. 

To date, there have been several systematic reviews published on probiotic use in elderly with diagnosed conditions and diseases [38,39], but few have specifically focused on their effects in healthy elderly. Although it is important to investigate the use of probiotics to alleviate aging-associated diseases and conditions, it is also important to evaluate their use to confer health benefits in elderly without a specific diagnosis. Furthermore, the systematic reviews on the use of probiotics in healthy elderly have focused only on specific health outcomes, such as immune function, constipation, diarrhea etc. [40,41,42,43], and a comprehensive overview of the effect on multiple health outcomes is lacking. As there have been a number of clinical studies utilizing a wide array of different probiotic strains with conflicting results, it is necessary to better understand the effects probiotics have on microbiota composition, gastrointestinal symptoms, immune function, etc. in healthy elderly. Therefore, this systematic review will provide a comprehensive and unbiased summary of randomized, controlled clinical trials with probiotic supplementation, evaluating which outcomes have been examined and what has been the effect of these interventions. 

## 2. Methods

### 2.1. Protocols and Registration

This systematic review is registered in PROSPERO under ID CRD42021231422. Details of the systematic review were initially submitted to PROSPERO on 9 March 2021. To enable PROSPERO to focus on COVID-19 registrations during the 2020 pandemic, the registration record was automatically published exactly as submitted and not checked for eligibility. The record was formally registered on 8 April 2021. 

### 2.2. Information Sources and Search Strategy

A comprehensive search was conducted by a medical research librarian in the following electronic bibliographic databases: PubMed, Embase, Cochrane Central Register of Controlled Trials (CENTRAL), Cinahl and Scopus. Briefly, the search strategy was built up by the MeSh term Probiotics combined with Probiotic* using the Boolean operator OR; the following limits were set: English, Randomized Controlled Trials, Aged or Elderly or Older adults. The complete search strategy applied in the different databases is available in the Appendix A (Appendix A). All records identified in the search were imported into an EndNote library and duplicates were removed. 

### 2.3. Eligibility Criteria

All clinical randomized controlled trials published in English were eligible for inclusion in that they reported interventions with probiotic and synbiotic supplements in healthy older adults (defined as 60 years and older). Studies reporting clinical trials involving certain patient groups or otherwise unhealthy participants, as well as vaccine studies, were excluded. Grey literatures, such as theses or commentary articles, were excluded. 

### 2.4. Study Selection

Two researchers (SA and KK) independently screened all the titles and abstracts of the unique records in the search results and the results from this screening was then checked by a third researcher (LT). Discrepancies were discussed between these three reviewers. Full texts of potentially eligible studies were retrieved and independently assessed for eligibility by five reviewers (AH, CB, KK, MS, JA). When the eligibility of a particular study was unclear, this was discussed with other members of the review team. 

### 2.5. Data Extraction 

Data extraction was done using the Cochrane Data collection form for intervention reviews: RCTs only (CDPLPG Version 3, April 2014). The work was divided between five authors (AH, CB, KK, MS, and JA), and after the initial extraction all extracted information was compiled in a master extraction matrix. The first author (AH) was responsible for quality checking the added information for all studies. When the extraction matrix was complete for all included papers, the authors familiarized themselves with the complete data set, and the results were then discussed between all authors in a digital meeting where discrepancies were identified and corrected. 

An updated search was performed by a librarian in April 2021, which resulted in one more study being included in the final analysis. The abstracts from the updated search were screened and reviewed by two independent researchers (CB and AH) and quality checked accordingly.

### 2.6. Risk of Bias and Study Quality Assessment

The quality of the included studies was assessed both using the Cochrane Data collection form for intervention reviews: RCTs only (CDPLPG Version 3, April 2014) incorporating relevant questions for randomized controlled studies and further using a checklist that emphasized some particular questions. In brief, the checklist considered the clarity of the research questions and aims, the clarity of the description of the methods and results, and the risk of bias in the laboratory and statistical methods. Assessments were performed for each study by two reviewers of the review team (AH, CB), and disagreements between the reviewers were resolved through discussion with the rest of the review team. The full version of the quality checklist can be found in the Appendix A (Appendix A). The heterogeneity among the included studies, e.g., in methods and design, prohibited any meta-analyses of the material. Instead, a narrative analysis was conducted for each outcome listed in Table 1.

## 3. Results

This review includes 17 randomized placebo-controlled trials studying different outcomes of probiotic treatment in healthy older adults [44,45,46,47,48,49,50,51,52,53,54,55,56,57,58,59,60,61]. Our search conducted in December 2019 returned 1659 entries of which 16 were ultimately included in this review [44,45,46,47,48,49,50,51,52,53,54,55,56,57,58,59,60]. An updated search of the literature search in April 2021 returned one further article to be included in our final manuscript [61]. Refer to Figure 1 for an overview of the search strategy. The included studies were highly varied in their aims, outcomes, design and methods. Broadly speaking, the outcomes of the studies fell into one or more of the following categories: (a) investigation of changes in the gut microbiota [44,47,48,49,50,52,54,56,58,59,61]; (b) immune-related measurements [44,46,47,50,52,53,54,55,56,58,59]; (c) digestive health [44,50,51,54,56,57,58,61]; (d) general well-being and cognitive function [44,50,51,57,58,61], and (e) lipids and other biomarkers [45,47,61]. Some of the studies sought to determine the effects of single strain probiotics [49,52,53,55,58], others of mixtures of different probiotics [44,45,46,51,54], or probiotics in combination with prebiotic sources, i.e., synbiotics [47,48,49,50,56,57,59]. In terms of laboratory methods, a wide range of different methods were used, such as PCR (real-time [44,56] and qPCR [47,48,54,56], culture-based methods [48,49,52,59], FISH [50,58], flow cytometry [56], RAPD-DNA [52] and sequencing [47,54,61] to explore changes in gut microbial communities. qPCR was also used for immune markers together with various immunoassays and flow cytometry [46,47,53,54,58]. In addition, stimulation of immune cells was used in a few studies to look further into cytokine secretion and changes in different immune cell populations after LPS [50,58,59] or PHA [52,54] stimulation. The number of study participants ranged from 18 to 1072, and the studies originated from 12 different countries. Refer to Table 1 for an overview.

### 3.1. Assessment of Study Quality

The risk of bias assessment included the following domains: random sequence generation, allocation concealment, blinding of participants and personnel, blinding of outcome assessment, incomplete outcome data, selective outcome reporting, and other bias. All included studies were judged by two independent researchers from the team (AH, CB) according to these sections, and the overall risk of bias was considered low. However, many studies were unclear in reporting sections with descriptions of the random sequence generation [45,49,50,52], allocation concealment [45,47,49,52] and blinding [47,49,54,55,56] potentially introducing selection, performance bias as well as detection bias respectively. Risk of bias due to incomplete outcome data was reported for one study [45]. However, none of the included studies were judged as having a particularly high risk of bias.

Furthermore, there was an adequate level of detail in the information on study design, research questions, aims and inclusion of participants as evaluated from the checklist. However, a major limitation for most studies was the small number of subjects included. Only eight of the studies reported a proper sample size calculation. The description of the laboratory analyses was generally sufficient, although some key details were either unclear or not reported in many studies such that replication of the methods would be difficult. Details regarding the processing and statistical analysis of the data was also highly variable on the individual level. The included papers were scored from our quality checklist from 0–3, with a mean score of 2.81 (range 2.63–3.0) (Appendix A).

### 3.2. Microbiota Composition

Of the 17 studies included in the review, eleven studies assessed the impact of either probiotic or synbiotic consumption on gut microbiota composition (See Table 2 for overview of probiotic strains, methods etc.) Of the eleven studies, six studies were probiotic [44,49,52,54,58,61] and five were synbiotic studies [47,48,50,56,59]. For four studies, the impact on microbiota was the primary outcome [48,49,50,51], whereas for the seven remaining studies, microbiota composition was a secondary outcome. All eleven studies evaluated fecal microbiota composition. Different methods of assessment were used with the majority of studies utilizing polymerase chain reaction (PCR): Six studies used qPCR [44,47,48,54,56,58]. Four studies used culture-based methods [48,49,52,59]. Two studies used fluorescent in situ hybridization (FISH) [50,58]. One study used flow cytometry [56], one study used RAPD-DNA [52], two studies used Illumina sequencing [47,61], and one study used pyrosequencing in addition to qPCR [54]. Ten of the eleven studies demonstrated some effect on microbiota composition. The study that failed to show an effect [59] used culture-based methods for assessment, with only six participants in the intervention and placebo groups respectively.

#### 3.2.1. Probiotics

Four single species probiotic interventions were performed [44,49,52,58], and one study examined multi-species interventions [54]. Two of the single-species studies assessed the impact of *Bifidobacterium* (B) species on fecal microbiota, both using the *B. lactis* HN019 strain and both showing presence of the supplemented bacteria at the end of the intervention [49,52]; in the study conducted by Arunachalam et al., the presence of the supplemented bacteria in stools post-intervention was the only investigated outcome in terms of gut microbial evaluation [52]. Ahmed et al., however, could also show significant increases in *Bifidobacterium* at the end of their four-weeks intervention, when compared to pre-intervention levels [49]. In addition, significant increases in *Lactobacillus* (L) and *F. Streptococci* were seen at medium and high doses of the probiotic supplement. The increase in *Lactobacillus* may, however, partly be attributed to the presence of lactose in the milk powder, as the placebo group also demonstrated a slight increase in this bacterial family. Additionally, statistically significant decreases were demonstrated in *fecal coliforms,* at high and medium doses of *B. lactis* HN019. However, no changes were seen in *total anaerobes* or *Bacteroides* counts, and mold and yeast showed such high inter-individual variations that the data was not further interpreted. The levels of *Bifidobacterium*, *Lactobacilli* and *Enterococci* remained higher compared to pre-intervention levels after a two-weeks wash-out period, although not statistically significant. 

Another single strain study looked at the effect of 28 days of *Bacillus coagulans* supplementation and found a significant increase in *Bacillus* spp. and *Eubacterium rectale* compared to baseline, however, neither species showed significant effects when compared to placebo [58]. Interestingly, *Fecalibacterium (F.) prautznitzi* species demonstrated a significant effect after 28 days when compared to placebo. In addition, Guillemard et al. evaluated the effects of *Lactobacillus casei (L. caracasei)*; the probiotic was however delivered in a fermented milk product and, hence, also containing the yoghurt symbiotic strains [44]. The study detected significantly more *L. casei* in the treatment group’s stool following one, two, and three months of the intervention. One month after the intervention, there was, however, no longer any difference in *L. casei* levels compared to baseline or between the groups.

Furthermore, two studies used supplements involving more than one probiotic strain [54,61]. Spaiser et al. evaluated the effect of *L. gasseri ICS-13*, *B. bifidum G9-1* and *B. longum MM-2* strains in a 3-weeks intervention [54]. This study found significant increases in *Lactobacillus* and *Bifidobacteria*, together with significant decreases in *E. coli* after 3 weeks. No microbiota composition pattern changes were observed (Unifrac-based analysis of changes). There were, however, more OTUs increased in the probiotic group, suggesting OTU enrichment, and it was noted that the OTUs close to *F. prausnitzii* increased during the intervention period. Kim et al. investigated the effects of a 12-week intervention with *B. bifidum* BGN4 and *B. longum* BORI on gut microbiota composition; using 16S rRNA sequencing, they found that at the genus level, there were significant changes in the probiotic group compared to control [61]. At 12 weeks, the probiotic group resulted in decreases in the relative abundances of *Eubacterium, Allisonella*, Clostridiales, and Prevotellaceae. 

#### 3.2.2. Synbiotics

Three studies evaluated the effect of *Bifidobacterium* probiotic species in combination with prebiotics on microbiota composition [48,50,59]. Two of these studies combined the probiotic supplement with Raftilose Synergy 1 (chicory inulin and oligofructose) [48,50]. Bartosh et al. combined Synergy 1 with *B. lactis BL-01*, *B. bifidum BB-02* [51] while Macfarlane et al. used *B. longum* [48]. In this study, significantly higher total *Bifidobacteria* counts (agar plate cultures) were observed in the synbiotic group compared to placebo during feeding (week four) and post-feeding (week eight), and qPCR showed significantly higher *B. bifidum* in the synbiotic group at week four. Significantly higher *Lactobacillus* counts were also observed during feeding (week four), when compared to placebo; however, this was observed without any difference in the specific strains assessed by PCR. No significant differences were observed in total fecal anaerobes. Similarly, supplementation with Synergy 1 and *B. longum*, as studied by Macfarlane et al., showed significant increases in total *Bifidobacterial* populations at week two and four and specific increases in *B. angulatum* and *B. longum* at both time points when compared to placebo [50]. Additionally, *B. adolescentis* and *B. bifidum* where significantly increased at week four. On a phyla level, Actinobacteria and Firmicutes were significantly higher during synbiotic consumption, and Proteobacteria were reduced by the synbiotic and differed significantly from the placebo at two and four weeks. At baseline in the synbiotic group, Firmicutes and Bacteroidetes were the most abundant phyla, whereas Firmicutes and Actinobacteria were most prevalent after the synbiotic intervention period. In the synbiotic group, the Firmicutes/Bacteroidetes ratio increased from 1.3 (baseline) to 6.6 (four weeks), with no differences found in the placebo group. The third *Bifidobacterium* study investigated the effect of 4 weeks of *Bifidobacterium animalis ssp. lactis BB-12* in combination with yacon (prebiotic source) and soy extracts on microbiota [59]. However, no differences in *Bifidobacterium*, *Clostridium* or *Enterobacteria* cultures were noted between synbiotic or probiotic groups, nor at any specific time point compared to baseline. 

Supplementation with *Lactobacillus* species and prebiotics was studied in two trials [47,56]. Supplementation with *L. acidophilus* NCFM together with Lactitol resulted in a difference between time points in the total numbers of microbes in the synbiotic group [56].*Bifidobacteria* were significantly increased in the synbiotic group, and there was a significant difference in *L. acidophilus* NCFM counts between the synbiotic and placebo. Significant increases were also noted in sulphate-reducers between intervention and washout period in both groups. No effects were seen on *Clostridium perfringens. Lactobacillus* (*Lactobacillus rhamnosus GG* and *pilus-deficient L. rhamnosus GG-PB12*) in combination with Promitor™ Soluble Corn Fiber (SCF, a candidate prebiotic) were evaluated in a cross-over design (*L. rhamnosus GG* + SCF, *L. rhamnosus GG*-PB12 + SCF, SCF only, placebo) by Costabile et al. [47]. The presence of both probiotic strains were confirmed during the synbiotic intervention period by qPCR (16s rRNA), and Illumina sequencing, and profiling (16s rRNA variable region V3,V4 sequencing using primers) was done to assess impact on microbiota composition. Phylogenetic analysis showed that the three interventions (*L. rhamnosus GG* + SCF, *L. rhamnosus GG*-PB12 + SCF, SCF only) significantly altered microbiota composition with a 2% variation in composition from baseline in each intervention group. Principal component analysis showed increased abundance of *Parabacteroides* and *Ruminococcaceae*. Significant increases in *Parabacteroides* were seen when compared to placebo in *L. rhamnosus GG* +SCF (4.3% increase) and *L. rhamnosus GG-PB12* + SCF (3.4% increase). *Ruminococcaceae incertae sedis* was significantly increased in the *L. rhamnosus GG* +SCF (2.4% increase) and SCF only (2.4% increase) arms. Both the symbiotic supplemented arms demonstrated significant decreases in *Desulfovibrio: L. rhamnosus GG* +SCF (0.09% decrease), *L. rhamnosus GG-PB12* + SCF (0.1% decrease). The *L. rhamnosus GG* +SCF arm showed a significant decrease in *Oscillospira* (0.04% decrease).

### 3.3. Immune-Related Measurements

The effect of probiotic supplementation on markers of humoral immunity were investigated in seven studies. Three studies presented results derived from serum [46,47,53], while the other four utilized stimulation of cell cultures by either phytohemagglutinin (PHA) [52,54], lipo-polysaccharide (LPS) [58] or both [50]. Moreover, six of the studies (46,47,52,53,54,58) expand on their humoral immunity findings with cell-mediated immunity observations, i.e., investigating treatment effects on immune cell activity and population changes. In addition, two studies investigated the efficacy of probiotic supplementation to decrease the incidence and/or duration of common cold [55] or other common infectious diseases (CIT) [44]. Finally, three studies examined fecal immune markers as an outcome [56,58,59]. 

#### 3.3.1. Markers of Humoral Immunity 

Moro-Gacria et al. examined the effects of a six-month intervention with *Lactobacillus delbrueckii* spp. *bulgaricus* 8481 on serum markers of humoral immunity [53]. The probiotic treatment significantly increased the levels of human b-defensin-2 (hBD2) at both three and six months. A range of different interleukins and other inflammatory markers were also measured (IFN-γ, IL-1β, IL-2, IL-4, IL-5, IL-6, IL-8, IL-10, IL-12p70, TNF-α and TNF-β); the only significant finding was a decrease in IL-8 in the probiotic group compared to baseline at six months. Contradicting these findings in a different study, *B. longum* Bar33 and *L. helveticus* Bar13 intervention for 30 days significantly increased serum levels of IL-8 in the probiotic group compared to both baseline and placebo [46]. In addition, TNF-α levels decreased after probiotic treatment, while no changes were observed in immunoglobulins (IgA, IgG), IL-6 or CRP. In a crossover study, the changes in IL-6, IL-8 and CRP serum levels were examined in response to four treatments: Soluble Corn Fiber (SCF) as prebiotic, SCF in combination with *Lactobacillus rhamnosus* GG, SCF in combination with *Lactobacillus rhamnosus* GG-PB12 (pilus-deficient derivative), or placebo for 3 weeks [47]. The SCF prebiotic alone significantly decreased the levels of IL-6 compared to baseline, and the GG-PB12 + SCF treatment resulted in decreased CRP levels compared to baseline. No other immunity-related differences were found. 

In addition to investigating levels of serum immune markers as an outcome, four studies also examined stimulation experiments. A 3-week crossover intervention with a multi-strain probiotic (consisting of *L. gasseri* KS-13, *B. bifidum* G9-1, and *B. longum* MM-2) found that IL-5 and IL-10 production from cultures of peripheral blood mononuclear cells (PBMCs) stimulated with PHA increased in the probiotic group [54]. Consistent with this finding, PHA-stimulated PBMCs showed a significant increase in anti-inflammatory IFN-α secretion after 6 weeks of *B lactis* HN019 consumption (both compared to baseline and to placebo) in the study by Aranachalam et al. [52]. In addition to the two studies using PHA stimulation, two trials examined LPS-stimulated PBMCs. LPS-stimulated PBMCs showed no differences in their secretion of IL-6, IL-10, TNF-α and IL-1β when comparing *Bacillus coagulans* GBI-30 supplementation to placebo [58]. However, the probiotic-treated group showed a significant increase in IL-10 production compared to baseline after the 28-days intervention, and TNF-α levels in the placebo group increased . The plasma levels of CRP showed no significant changes in either group. Furthermore, LPS-stimulated whole blood showed that supplementation with *B. longum* and an inulin-based prebiotic resulted in significant decreases in TNF-α compared with placebo both at two and four weeks [50]. Concentrations of MCP-1, IL-6, and IL-8 were reduced at two weeks in the synbiotic group, but there were no differences between groups. Declining trends were also observed for IFN-γ and IL-4 after four weeks of treatment. In addition, this study looked at serum levels of CRP, IgG and IgA without finding any significant changes. 

#### 3.3.2. Immune Cell Activity and Population Changes 

Six studies (46,47,52,53,54,58) used different methods to study innate and adaptive immunity, such as NK cell and T cell activity and examined different immune cell population changes (refer to Table 1). The NK cell activity reflects the capacity of the effector cells (NK cells) to phagocytize target cells [46,47,52,58]. The studies differ in their design including a range of different innate and adaptive immune cell populations examined, as well as various E/T (effector to target) ratios and target cells (human malignant or bacterial). 

Six weeks of supplementation of *B. lactis* (HN019) with milk significantly impacted NK cell activity [52]. The phagocytic capacity tripled in the probiotic group, which was a significant increase both within group and compared to placebo. The higher phagocytic capacity also remained significantly elevated at the follow-up after six weeks (E/T ratio of 40:1, using *Staphylococcus aureus*). Additionally, the bactericidal activity (reported as a percentage of phagocytized bacteria) increased significantly from baseline in both groups at three and six weeks of supplementation, as well as at the six-week follow-up. In contrast, *Bacillus coagulans* BI-30 treatment for 28 days resulted in no significant changes in NK cell activity [58]. Similarly, supplementation with Soluble Corn Fiber (SCF) alone or in combination with one of two probiotic supplements (SCF with *L. rhamnosus* GG or its pilus-deficient derivate *L. rhamnosus* GG-PB12) failed to elicit any significant changes in NK cell cytotoxicity toward K562 [47]. Interestingly, male and female participants differed significantly in NK cell activity at baseline; women had lower activity. In line with this observation, a trend toward greater improvement in NK cell activity was observed in the female group (*p* = 0.064) after SCF + GG supplementation. Additionally, the combination of SCF + GG increased NK cell activity significantly in subjects 70+ years old (as opposed to 60+). No changes in T lymphocytes, NKT cells and NK cells (CD56^dim^ and CD56^bright^ separately) were found in this study. However, in a study investigating one-month supplementation with *B. longum* Bar33 and *L. helveticus* Bar13, there were significant increases in B and Treg lymphocytes both within and between the placebo and probiotic groups [46]. No significant changes were seen in total T helper cells (CD4+) or T cytotoxic cells (CD8+), but their naïve subpopulations increased significantly, while CD4+ effector memory cells decreased and the CD8+ activated memory cells increased. A significant increase in NK cell ability to induce apoptosis in K562 cells at E/T ratio of 50:1 was also observed. Cytotoxicity increased to 73%, while placebo remained at the baseline value of 41%; however, this was observed without changes in NK cell population counts. Furthermore, although the probiotic combination of *L. gasseri* KS-13, *B. bifidum* G9-1 and *B. longum* MM-2 consumed over three weeks had an effect on humoral immunity (as described in the section above), a significant decrease in the CD3+ lymphocyte populations were observed in the placebo group only as well as a decrease in CD4+ cells (observed only in the first period of the cross-over study) [54].

The most extensive study of immune cell population changes was reported in the study by Moro-Garcia et al. [53]. They investigated if supplication with *L. delbrueckii* spp. *bulgaricus,* 8481 and *S. thermophilus*, 8357 for six months would have an effect on T cell population. No changes were found in the placebo group for any of the investigated parameters. However, in the probiotic group, CD8+ lymphocytes decreased significantly at the three-months follow-up, and the CD4+/CD8+ ratio significantly declined (without any significant changes in the CD 4+ population). In addition, the NK cell count increased significantly both at three and six months, and the senescence-related CD8 + CD28^null^ cell numbers significantly declined at both time points. Furthermore, maturity of the CD4+ and CD8+ cells was studied by investigating naïve, central memory (CM), effector memory (EM) and effector memory cells re-expressing the CD45RA (EMRA), focusing on the change from baseline to six months. Among the CD4+ cell population, levels of naïve cells and EMRA cells increased, while CM cells and EM cells decreased. Among the CD8+ cells, levels of naïve cells and EMRA cells increased significantly, while EM cells decreased. The EMRA cells were further investigated on basis of their level of differentiation to pE1, pE2 and pE. In both CD4+ and CD8+ cells, the least differentiated pE1 cells increased, while the most differentiated pE decreased. Finally, this study also investigated CD4+ and CD8+ T lymphocyte proximity to the thymus using the T-cell receptor excision circles analysis (TREC); which gives a further indication of the number of the most recent naïve T cells. The immature T cells CD31 + CD45RA showed a significant increase in both CD4+ and the CD8+ subsets after six months of treatment. However, all treatment effects had r disappeared by the time of follow-up six months after treatment cessation.

#### 3.3.3. Common Cold and Other Infection Episodes

Incidence and duration of the common cold and other infection episodes were studied in two trials [44,55]. In a population pre-selected for relatively low IgA saliva levels, heat-killed *L. pentosus* b240 was administered for 20 weeks in low and high doses and the incidence of the common cold was evaluated [55]. Although the mean duration showed no differences between the groups, the incidence presented a dose-dependent declining trend with the high-dose group having a significantly lower incidence rate. Similar results were found for the accumulated incidence rate with a significant decrease between groups and a significantly decreased accumulated incidence in the high-dose group. The incidence and duration of common infectious diseases (CID) and their subcategories were further investigated in a study that examined the effects of supplementation with a sweetened fermented dairy product containing *L. casei* DN-114001 combined with *S. thermophilus* and *L. delbrueckii* spp. *bulgaricus* [44]. While no significant changes were found in the CID incidence, the episode duration and accumulative duration were significantly lower in the treatment group as compared to the placebo. As opposed to the gastrointestinal tract infections and the lower respiratory tract infections, a significant decrease in duration of upper lower respiratory tract infections (ULRT) was also shown (both per episode and accumulated). In addition, rhinopharyngitis—a ULRT subcategory—showed a significant decrease in both per episode duration and accumulated duration.

#### 3.3.4. Fecal Immune Function

Three studies investigated fecal immune parameters [56,58,59]. One study co-cultured distinct murine macrophage cell lines with the soluble fraction of feces stimulated with LPS to assess the effects on produced cytokine levels [59], while two studies determined the levels of biomarkers directly from the fecal samples [56,58]. In the murine macrophage model (RW364 cells), administration of soy and yacon product with *B. animalis* spp. *Lactis* BB-12 for four weeks resulted in no significant changes in cytokine production levels (IL-6, TNF-α and IL-10) [59]. In the two studies measuring biomarkers directly from the fecal samples, a significant increase of PGE2 levels after two weeks of *L. acidophilus* NCFM + lactitol consumption was found between treatment groups [56]. A significant increase in IgA levels two weeks after treatment cessation was also found within the probiotic group, although there were no differences between groups. Furthermore, this study also showed a trend (*p* = 0.0821) for reduced calprotectin levels in the synbiotic group compared to placebo. In contrast, 28 days of *Bacillus coagulans* GBI-30 administration had no effect on calprotectin levels [58].

### 3.4. Digestive Health

Digestive health was studied in eight [44,50,51,54,56,57,58,61] trials using different probiotic strains and varying length of intervention. Of these, only one examined the number of gastrointestinal infections and reported no statistical difference between groups after three-months consumption of a fermented dairy product containing *L. casei* DN-114001 [44]. Gastrointestinal symptoms were examined as an outcome in seven studies [50,51,54,56,57,58,61]. Of these, four studies reported some significant improvements including: symptoms of abdominal pain after consumption of *B longum* in combination with a prebiotic mixture of inulin and oligofructose during four weeks [50]; improved defecation frequency, bowel movement frequency and stool frequency after ingestion of a mixture of Bifidobacterium strains *(B. longum BB536, B. breve M-16V, B. infantis M-63 and B. breve B-3)* in combination with moderate resistance training for 12 weeks [51]; improved stool frequency after a two-week intervention with *L. acidophilus* NCFM in combination with lactitol [56]*;* and frequency of gas passage and abdominal distension was significantly improved after 12-weeks of supplementation with *B. bifidum* BGN4 *and B. longum* BOR1 [61]. Finally, four studies reported no significant effects on bowel habits: no effects were reported from interventions with *B. longum* in combination with a prebiotic mixture of inulin and oligofructose during four weeks [50]; *L. reuteri* for 12 weeks [57]; *Bacillus coagulans* BC30, 6086 for four weeks [58]; or following a three-week intervention with a probiotic mixture of *L. gasseri* KS-13, *B. longum* MM-2 [54]. 

### 3.5. General Well-Being and Cognitive Function

General well-being was included as an outcome in six trials with probiotic supplementation [44,50,51,57,58,61], but only one of these studies could report any statistically significant differences in well-being, quality of life or perceived stress after the interventions [61]. Kim et al. reported that 12-weeks supplementation with *B. bifidum* BGN4 and *B. longum* BOR1 did affect the stress score as measured with a validated 20-item self-reported questionnaire. While the stress score was increased in the placebo group, it was statistically significantly decreased in the probiotics group. However, quality of life scores (measured with the Satisfaction with Life Scale: SWLS) and depression (measured with The Korean version of the Geriatric Depression Scale: GDS-K) or Positive Affect and Negative Affect Schedule (PANAS) scores did not change significantly. Lower stress levels were also reported in one study in subjects suffering from indigestion or abdominal pain, evaluated by the Perceived Stress Scale (PSS) [57]. Another study reported improved mental state assessed with the Patient Health Questionnaire-9 (PHQ-9) and the Generalized Anxiety Disorder-7 (GAD-7) questionnaires, in the probiotic group [51], while two studies reported no general effect on mental health [57,58]. 

Cognitive function was assessed as the primary outcome in two studies. Inoue et el. used the Japanese version of the Montreal Cognitive Assessment instrument (MoCA-J) to assess cognitive function and measured executive function and inhibitory controls using a flanker task [51]. A flanker task is a set of response inhibition tests comprising two types of stimuli, the central target letter of which is flanked by noise letters. No significant effects were observed between probiotic and placebo in the MoCA-J scores. However, flanker task scores increased more significantly in the probiotic group compared to placebo. The other study by Kim et al. used CERAD-K, a validated cognitive test battery that scores language, memory, visual-spatial processing, and attention/executive function [61]. They found that scores of mental flexibility had improved significantly at week 12 compared to placebo (active treatment contained *B. bifidum* BGN4 and *B. longum* BOR1); however, the other domains of the battery did not differ significantly between groups. 

### 3.6. Lipids and Other Biomarkers

Blood lipid profile was included as an outcome by Costabile et al. [47]; this study found reduced total cholesterol and LDL-cholesterol in volunteers with initially elevated concentrations (TC > 5 mmol/L) at baseline (*n* = 26) after a synbiotic intervention with *L. rhamnosus GG* combined with SCF (but not with *L. rhamnosus GG-PB12*). Glucose levels were also examined without significance differences between groups. Other biomarkers such as serum calcium levels were shown by Gohel et al. to be improved significantly in the probiotic (*L. helveticus MTCC 5463*) group with mean (SD), while the placebo group had a significant decrease of calcium levels [45]. There were, however, no significant effects observed for hemoglobin or hematological parameters (total leukocytes count, red blood cell count, Mean Corpuscular Volume (MCV), platelet count or erythrocyte sedimentation rate). Another study performed by Kim et al. reported improved brain function evaluated as levels of BDNF in blood with probiotic treatment [61]. Serum BDNF levels were significantly increased at week 12 in the probiotics group (*B. bifidum* BGN4 and *B. longum* BOR1) compared to the placebo.

## 4. Discussion

As revealed in this systematic review, the main outcomes hitherto examined in randomized, placebo-controlled trials with probiotic supplementation in healthy older adults included gut microbiota composition, immune function (including markers of humoral immunity, changes in immune cell population and function, incidence of infection, etc.), digestive health (gastrointestinal symptoms, frequency, etc.), general well-being, cognitive function, and changes in lipids and other biomarkers (Table 1). This narrative overview showed that the efficacy of probiotics/synbiotics to positively affect these outcomes was highly variable; however, the findings from the included studies suggest that probiotics can influence gut microbiota composition in healthy elderly, positively impact age-related immune dysfunction, and potentially have moderate effects on a wide variety of health outcomes. 

### 4.1. The Effects of Probiotic Use as an Intervention in Healthy Elderly

#### 4.1.1. Microbiota Composition

Gut microbiota composition was the most commonly assessed outcome in the included studies [44,47,48,49,50,52,54,56,58,59,61], although only four stated it as the primary outcome [48,49,50,61]. We find it quite surprising that so few of the reviewed studies utilized sequencing technologies, i.e., next generation sequencing, for microbiota analysis [47,61]. At this point, sequencing technologies have been used for over a decade to elucidate gene content of the gut microbiome and is nowadays the primary choice for many gut microbiome studies, as it has an unprecedented coverage and evades some of the more classical methodological difficulties. We speculate that this is, at least in part, due to the relative high cost of the method as compared to other available approaches. Although the sequencing costs have been extensively reduced since the method was first introduced, it is still a rather expensive method; especially considering that the data analysis requires bioinformatic competences which is not always readily available and may have to be paid for in addition to the sequencing itself. 

The most widely used method for sequencing the fecal microbiome thus far is 16S rRNA sequencing [62]. This sequencing method utilizes PCR to target and amplify portions of the hypervariable regions (V1–V9) of the bacterial 16S ribosomal RNA subunit gene. In this way the method can differentiate between different bacterial organisms, as the 16S ribosomal RNA subunit gene contains two regions that are conserved throughout all bacterial species while the V1-V9 regions are unique for each genus [62]. Both the studies reviewed here [47,61], performed 16S rRNA sequencing of the hyper variable regions V3-V4 on the Illumina Miseq platform, but their methodological approaches diverged thereafter: Costabile et al. utilized the mare package in R (Korpela 2016 mare: Microbiota Analysis in R Easily. R package version 1.0), performing taxonomic annotation of the reads using USEARCH [47], whereas Kim et al. processed their microbial sequences using QIIME2 and aligned representative OTU sequences based on the SILVA database [61]. Although using a well-known, high-quality sequencing platform and validated approaches for downstream analysis of the reads, detailed description of the whole sequencing pipeline is key, as apparently small differences in the strategy of analysis may produce large differences in the down-stream interpretations [63,64,65]. Going forward, as studies utilizing sequencing start to accumulate in this field, we advise on the inclusion of a detailed description of sequencing methodologies and analyses strategies that, preferably, have been previously used within the field to increase validity, reproducibility and comparability of future findings. However, it is important to also examine pitfalls of classical approaches and to update methodological choices when necessary, as laid out by Knight et al. in their publication “*Best practices for analysing microbiomes*” [66].

Furthermore, the two methods used most frequently for investigating gut microbial changes, as reviewed here, were qPCR and culturing. The weakness of both methods is the relatively low number of bacterial organisms that can be evaluated, especially in contrast to 16S rRNA sequencing where the numbers are practically infinite. In contrast to culturing, which is a more semi-quantitative approach to bacterial cell quantification, qPCR is a highly sensitive measurement that allows for absolute quantification (when a standard curve is used). In favor of bacterial culturing, only live organisms will be culturable with this method, and most often it is the effect of *live* probiotic bacteria these trials aim to investigate (if the bacteria is not heat-killed prior to consumption as in the study of Shinkai et al. [55]). Furthermore, in a couple of studies reporting on findings from qPCR, the methodological description had some weaknesses, e.g., in nomenclature. We advise that future studies refer to The MIQE Guidelines [67], to make sure that the minimum requirements of information for publication of quantitative Real-Time PCR experiments are met.

Six of the studies used different species of *Bifidobacterial* in their supplements, while three used only *Lactobacilli*, one used a combination of the two and one used *Bacillus coagulance* (genus *Bacillus*) (see Table 2 for overview). Nevertheless, the most common changes in gut microbial communities found between treatment and placebo were in the *Bifidobacterium* genus. Noteworthy, is also the fact that all studies but one showed significant differences between the treatment groups, suggesting that probiotic and synbiotic supplementation should be regarded as an effective strategy to elicit changes in the gut microbiota of the elderly; bearing in mind of course, that two of the studies only made the effort to investigate the presence of the bacterial strain(s) that were supplemented. The study that failed to show any significant effects on microbiota composition after the intervention was performed by Manzoni et al. [59] and used culture-based methods for the microbiota analysis. The study had a small samples size with only six participants in the intervention and placebo groups respectively, rendering it quite under-powered; this might explain the absence of significant findings. The authors also speculate that high initial levels of *Bifidobacterial* numbers may have weaken their probiotic intervention, which utilized two *Bifidobacterial* species. 

Interestingly, *F. prautznitzi* increased significantly after 28 days-consumption of *Bac. coagulans GBI-30* when compared to placebo [58], and Spaiser et al. found that several bacterial groups matching *F. prausnitzii* where more prevalent in the probiotic group after the intervention (*L. gasseri KS-13, B. bifidum G9-1, B. longum MM2*), as compared to placebo [54]. *Fecalibacterium prautznitzi* has a particular place in the spotlight among the beneficial commensals as being a butyrate-producing bacteria with anti-inflammatory properties [68,69] that tend to decrease or be depleted in various diseases, such as inflammatory bowel diseases [70], colorectal cancer [71] and non-alcoholic fatty liver disease [72], and last but not least in aged individuals [73]. Hence, this is a potentially significant finding for promoting health maintenance in older adults, as mediated via the gut microflora. Nyangale et al. also found a significant increase of the anti-inflammatory cytokine IL-10 in the probiotic group over the treatment period (however, not significantly different from placebo) [58]. Notably, in experimentally-induced colitis murine models, *F. prautznitzi* can upregulate regulatory T-cells and induce IL-10 release [68,74].

Methodological heterogeneity among the reviewed studies, such as different analysis methods, different levels of microbiota assessment (i.e., group/genus/species specific probe sets) and variations in results reporting of microbiota values (eg CFU/g, PCR equivalent CFU or cells/mL), currently poses a challenge to the field. Hence, like others before us, we identified a need for greater standardisation and clearer reporting of methodological choices. 

#### 4.1.2. Immune-Related Measurements

An increasing number of studies suggest that aging is associated with an increased prevalence of chronic low-grade inflammation, and this low-grade inflammation has been in part linked to imbalances in the gut microbiota [11,12]. Therefore, apart from changes in gut microbiota composition, immune-related measurements were the most investigated outcome in the included studies. 

All three studies that examined the effects of probiotic supplementation on markers of humoral immunity found effects on immune function, e.g., decreases in IL-8 [53], TNF-α [46], and CRP [47]. However, the results reported in the different studies are somewhat inconclusive as they show changes in opposite directions, as in the case for IL-8 [46,53]. Differences like these might, at least in part, be attributed to differences in the study length; the intervention conducted by Moro-Garcia for example lasted for six months [53], whereas the intervention by Finamore et al. only lasted for 30 days [46]. In addition, in some of the trials that included follow-up visits after the intervention revealed that the acute and more long-term effects of probiotic/symbiotic supplementation can be quite different. Taken together, this underscores the importance of considering intervention length when comparing the results between studies. 

In addition to demonstrating that probiotic supplementation can affect serum immune markers, several of the included studies showed that ex vivo stimulation supports a beneficial role for probiotic supplementation, e.g., increased production of anti-inflammatory cytokines such as IL-10 [54,58] and IFN-α [52], or decreased production of pro-inflammatory cytokines such as TNF-α [50]. We, however, recognize that many of the studies have a common problem in identifying differences between the placebo and probiotic groups but can show significant differences when examining within-group effects. This observation could potentially suggest that many probiotic interventions (due to intervention length, selection of strains, etc.) may not be robust enough to induce changes that withstand between-group comparisons. Alternatively, selection of the placebo could be another potential complicating factor; depending on the choice of placebo, the ingredients may induce changes in immune function that dilute the effects attributed to the probiotic. Notably, even though many authors state that their chosen placebo is absorbed already in the small intestine, this does not mean that it is without effects per se. It may even be that changes taking place in the small intestine have more pronounced effects on some immune parameters; particularly since this part of the intestine harbors specialized lymphatic tissue, such as Peyer’s patches, that may interact with both probiotic bacteria [75] and prebiotic fibers [76]. Even with these potential issues, taken together, the findings from the studies with immune cell stimulation experiments suggest that probiotics potentially increase the production of anti-inflammatory cytokines and may work to dampen the inflammatory immune response. 

Furthermore, probiotic supplementation seems to positively affect immnunosenescence, at least to some extent. The reviewed studies showed that probiotic supplementation can counteract the reduced naïve T cell production and memory T cell accumulation commonly associated with aging [46,77,78], and by increasing numbers of regulatory T cells and B cells [46]. Moro-Garcia et al. also demonstrated that probiotic supplementation increased less-differentiated T cell populations [53], in part counteracting age-associated increases in highly differentiated effector and memory T cells [79]. Furthermore, Moro-Garcia et al. found decreased CD8 + CD28^null^ cell numbers, which is significant as the accumulation of CD8 + CD28^null^ cells has been linked to reduced immune response to infection and immunization response in the elderly [80]. 

The impact of supplementation on NK cell activity was, however, less conclusive. Several studies investigated the impact of probiotic supplementation on NK cell activity and cytotoxicity. Three studies showed some effects on NK cell activity [46,52] or NK cell numbers [53], while two studies failed to show any NK cell effects [47,58]. These findings may suggest some effects of probiotics/synbiotics on NK cell populations; however, the results presented to date are far too inconclusive and more studies are warranted. 

In summary, the findings from immune related investigations in the included studies suggest that probiotic supplementation may be efficacious in increasing immune cell naivety and shifting cytokine production to a more anti-inflammatory profile. Increased naivety and reduced memory/effector cell populations would be extremely beneficial in the elderly, as it would improve response to infections; supported by the two studies that showed a positive impact of probiotic supplementation on the incidence of common cold [55] and decreased episode duration of CID [44]. These immunological benefits may also be helpful in improving vaccine response, although studies with this outcome were not included here.

#### 4.1.3. Digestive Health

Gastrointestinal problems are a widespread phenomenon among older adults [57] and may contribute to reduced quality of life in the elderly. Hence, digestive health should be considered an important outcome to study, especially in terms of increasing individual wellbeing. Nevertheless, only one of the reviewed studies had digestive health as its primary outcome [57]. Concerning the effect of probiotic interventions on digestive health in elderly, the findings are very inconclusive. Three studies reported some effects, i.e., less abdominal pain [50], positive effects on defecation frequency and bowel movement frequency/stool frequency [51,56]. However, the same number of studies reported no effect on bowel habits [50,54,57]. Additionally, in a few of the studies it remains unclear if the positive changes reported can actually be attributed to the probiotic supplementation or not. In the study by Ouwehand et al. [56] for example, the positive effects on defecation frequency and bowel movement frequency/stool frequency may in fact be due to the Lactitol given in combination with the probiotic; Lactitol is a medicament commonly used to improve bowel function [81,82]. Likewise, the effects found in the study by Inoue et al. may in fact be attributed to the exercise training given in combination with their probiotic treatment [51], as physical exercise improves gut health among older adults. Hence, the effect of probiotic treatment on digestive health in older adults remains to be further investigated. 

#### 4.1.4. General Well-Being and Cognitive Function

Several of the studies examined the effects of probiotics on general well-being [44,50,51,57,58]. Two of the studies found that probiotic supplementation positively impacted well-being, decreasing anxiety and depression scores [51] and decreasing anxiety in participants suffering from indigestion and abdominal pain [57]. However, the majority of studies found no effect on mental health or general well-being [44,50,57,58]. Taken together, these findings make it difficult to interpret if probiotics positively impact general well-being and mental health in elderly. Several studies have suggested that probiotics can affect depression and anxiety [83,84] and can even influence brain circuitry underlying emotional control in healthy adults [85,86]. Because of the potential role for probiotics in affecting general mood and well-being, it is necessary to further investigate the effects in elderly in more sufficiently powered studies. 

In addition to general well-being, two studies examined the effects of probiotics on cognition [51,61]. Inoue et al. combined probiotic treatment with moderate resistance training and had cognitive function as their primary outcome as investigated by a well-known questionnaire (MoCA-J) in combination with a flanker task [51]. No significant effects on the MoCA-J scores could be observed with the probiotic supplementation, as the scores increased in both groups. Increasing evidence suggests that gut microbiota composition may be modulated by physical exercise [87,88,89]; therefore, the improved cognition may be in part due to the effects of the exercise training rather than the probiotic supplement. These findings are rather significant for the field, as they support further investigation of physical exercise and the gut-muscle axis as important regulators of gut microbiota composition and health outcomes in elderly. Although the probiotic did not significantly affect MoCA-J scores, the flank.er task scores were higher in the probiotic group compared to placebo. The potential for probiotics to positively impact cognitive function was further supported by Kim et al., as they found that probiotic supplementation improved scores of mental flexibility. However, no other domains of the CERAD-K cognitive test battery were affected by probiotic supplementation. Several studies in mice indicate that modifying the gut microbiota via probiotics is sufficient to influence memory deficits and improve cognitive function [90,91,92], suggesting that it could also potentially impact cognition in elderly humans. The modest effects observed in the included studies highlight the difficulty to replicate positive findings from mouse studies in human clinical trials, a prevalent issue in probiotic interventions. 

#### 4.1.5. Lipids and Other Biomarkers

Current guidelines for CVD risk reduction are primarily focused on strategies to reduce concentrations of LDL-cholesterol, with a focus on “lower is better”. Observational studies suggest that individuals with hyperlipidemia have a risk of CVD that is three times that of the population with normal lipid status. A reduction in serum cholesterol is strongly associated with a reduction in CVD risk [93,94]. From a public health perspective, lifestyle modification, including dietary changes, is considered a first step in controlling and treating CVD risk factors [95]. To date, both experimental and clinical studies have suggested that probiotic supplementation may have beneficial effects on serum lipid profiles [96]. However, we found only one randomized placebo-controlled study that has investigated blood lipid profile as an outcome [47]. The positive results from Costabile et al. are in line with previous research suggesting synbiotics to have more benefit in patients with hypercholesterolemia than in individuals with normal lipid concentrations. In addition, reductions in TC and LDL in the elderly have been reported to be greater than those in younger individuals, potentially due to higher baseline values.

Moreover, serum calcium levels improved significantly after probiotic supplementation [45]. This is an interesting finding because loss of bone mass is a common problem among older individuals and especially among older women [97]. Supplements of calcium are therefore recommended for osteoporotic patients with low calcium intake/absorption [98]. Additionally, gut microbiota (GM) is more and more recognized as an important determinant of bone health and compelling evidence supporting that probiotics may improve bone health is starting to accumulate [99]; both from animal and human studies. In elderly postmenopausal women, probiotics even seem to reduce bone loss in a quite similar magnitude as observed with calcium + vitamin D supplements [99].

### 4.2. Study Quality of the Research Field and Methodological Considerations within Studies

The risk of bias for the included studies was relatively low, based on the determination of selection, performance, detection, attrition, and reporting bias. However, several of the studies lacked clear information pertaining to random sequence generation, allocation concealment, and blinding, allowing for potential selection, performance, and detection bias. Unless the studies are described as being open-label, then it is necessary that the randomization, allocation, and blinding procedures are clearly described; this is especially important to promote the reproducibility of the results. The studies had adequate information pertaining to the aim, research questions, study design, methodology, and exclusion/inclusion criteria. However, all but eight of the included studies lacked information pertaining to sample size calculations. As the majority of the studies had a small sample size (ranging from 18 to 1072 with a median of 47 participants), it is difficult to judge if the lack of significant positive findings was in part due to inadequate sample sizes. This highlights the need for more well-designed, sufficiently powered studies to investigate the effect of probiotics in healthy elderly. 

In addition to revealing some potential risks of bias, this systematic review also uncovered several methodological issues that should be considered for the design of future studies. First of all, the included studies significantly varied in duration (ranging from two weeks to six months, with a median of four weeks). Although the shorter interventions may not have been long enough to potential desired outcomes, interventions of longer durations also present several complicating factors. The risk of compliancy issues increases with study duration, and the changing seasons and weather introduce variability in the results (more time indoors in the autumn/winter, more activity and Vitamin D in the spring/summer, more infections in the winter, etc.). In addition, females tended to be overrepresented as participants in the included studies. This is rather important as some of the studies showed differences between the sexes; for example, Costabile et al. found that male and female participants had significantly different NK cell activity at baseline [47]. Therefore, sex-specific differences at baseline as well as treatment response must be evaluated in all studies. Furthermore, the included studies also had a large range of participant ages (ranging from 60 to 95 years). Costabile et al. found that the combination of SCF and *L. rhamnosus* GG had a tendency to increase NK cell activity compared to baseline in participants 70–80 years old compared to younger elderly [47], suggesting age-specific effects. It is well-established that the gut microbiota undergoes significant changes in composition and diversity in the elderly [13,34]. However, several studies have indicated that there are differences in the gut microbiota composition between younger elderly and centenarians [15,16] and that there is considerable variation in the microbiota profiles of older adults [14]. These findings support the idea that including a wide range of ages of older adults potentially introduces variation in the gut microbiota profile at baseline and supports including a smaller range in age and/or more in-depth microbiota characterization at baseline. 

### 4.3. Strengths and Limitations of This Systematic Review

The main strength of this review is its systematic approach to identifying and synthesizing all interventions studies with probiotics performed on healthy older adults. Our systematic approach also included an assessment of study quality and potential biases, using the well- structured and highly regarded Cochrane Data collection form for intervention reviews: RCTs only, which provides an important examination of methodological challenges. We were able to include a large number of RCTS, and the study quality was fairly high. The review covers a rather large research topic/area and includes a diverse set of studies, as an effect of our broad research question, which can be seen as another strength. In addition, in order to increase the homogeneity of the included studies, we only included studies with a placebo control. Three randomized studies conducted in healthy elderly utilized a diet group as the control and lacked a true placebo control; therefore, we did not include these studies [100,101,102]. In addition, Meance et al. and Matsumoto et al. also lacked placebo controls and were excluded [103,104]. By excluding these studies, we were able to focus on the outcomes reported from placebo-controlled trials exclusively, strengthening the validity and accuracy of our conclusions. 

At the same time, our review faces some limitations. Heterogeneity between the studies meant that we were only able to provide a narrative summary of the current evidence. An additional limitation is that we specified that only studies undertaken in healthy older adults were to be included, which posed a challenge for the studies performed in elderly care/nursing homes. The individuals residing in such homes are unlikely to be without health complications and are less likely to be fully operative. Furthermore, in many studies the actual health status of the participants in elderly care/nursing homes was not clearly described or further commented on. This led us to primarily focus on studies of “free-living”/independently living/community-dwelling older adults. However, one study was included when the participants were recruited from a nursing home since we found no grounds on which to exclude this study other than the context in which it was performed; the health status of the participants was fully investigated, described, and was consistent with our inclusion criteria [46]. In future reviews with a similar scope as this one, it could be beneficial to further clarify this point in the study protocol, as a number of studies have been performed to examine the effects of probiotics on older adults in elderly care/nursing homes [105,106,107,108]. 

### 4.4. Future Directions

With the growing evidence that the gut microbiome undergoes substantial shifts during the aging process [13,14,34], there is a need to better understand how modifying the gut microbiota via probiotic supplementation promotes healthy aging. In the present systematic review, we included studies that administered probiotics for a relatively short duration (median: four weeks) and assessed the effects directly after the treatment ended. It remains unclear if probiotics are more beneficial in preventing age-related disease and disability or if they are more efficacious in alleviating these conditions once they occur. In order to better understand how probiotics affect healthy aging, well-designed, sufficiently-powered longitudinal studies could be conducted in older middle-aged adults and elderly to assess how probiotics affect health outcomes after one, five, or ten years of treatment and then assess health outcomes for several years after treatment. Very few of the included studies assessed outcomes after a wash-out period; therefore, it would be interesting to establish how long the benefits of probiotics remain after supplementation has ended. Furthermore, as mentioned in an earlier section, a remaining question that could be addressed is if there is particular age window in which probiotic supplementation is particularly efficacious in promoting healthy aging. 

In addition, this systematic review uncovered several outcomes that need to be more thoroughly investigated in future studies. Only one study focused on the blood lipid profile as an outcome [47]; however, the results that *L. rhamnosus* GG combined with SCF was sufficient to reduce total cholesterol and LDL-cholesterol in participants with initially elevated concentrations was rather intriguing. Future studies designed to investigate the use of this synbiotic in preventing or treating high-cholesterol in elderly participants are warranted, especially since elevated cholesterols is a huge public health issue and seemingly small differences in blood lipids could have widespread positive effects on both individual and social parameters. Furthermore, although pre-clinical research show promise within this field, only two studies [51] focused on cognition as an outcome, however, the results were promising. We call for more in-depth examinations of the effects of probioitcs on brain function in healthy elderly, utilizing more extensive cognitive testing as well as functional magnetic resonance imaging (fMRI)-based studies. fMRI, both task-based and resting state, could be used to assess how probiotics alter brain connectivity in healthy elderly and how this connectivity correlates with other outcomes such as gut microbiota and immune profiles. 

## 5. Conclusions

Increasing evidence suggests that probiotic supplementation may be efficacious in counteracting age-related shifts in gut microbiota composition and diversity, thereby impacting health outcomes and promoting healthy aging. However, there is to date insufficient evidence to determine if a particular probiotic/synbiotic combination or duration of treatment is efficacious in improving health outcomes in healthy elderly. Probiotics can potentially have a modifying effect on gut microbiota composition and moderate effects on immune function, but the effect of probiotic supplementation on other health outcomes remains inconclusive. Caution is still needed when interpreting results from existing randomized, placebo-controlled trials, due to methodological concerns including limited sample sizes and subsequent low statistical power, suboptimal study designs, and variation in sequencing methods, statistical methods, and result interpretation. More robust research with well-designed and sufficiently powered studies are needed to further investigate if and the mechanisms by which probiotics can potentially impact healthy aging.

## Figures and Tables

**Figure 1 microorganisms-09-01344-f001:**
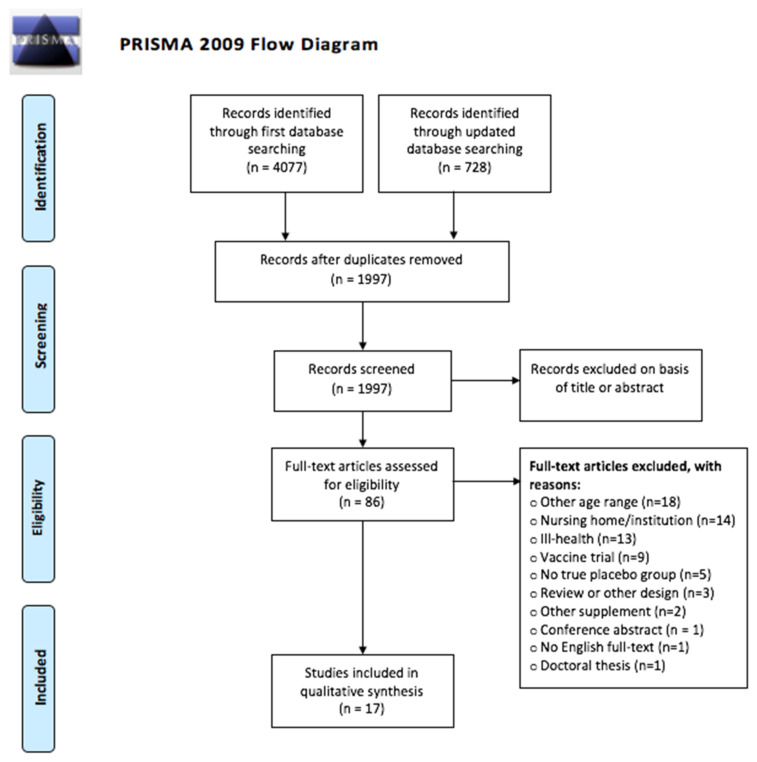
PRISMA Overview of article inclusion.

**Table 1 microorganisms-09-01344-t001:** Overview of included studies.

Study and Country	Design and Sample Size *	Sex (Female/Male)	Age ^#^	Probiotic/Synbiotic and Dosage in CFU/day	Duration of Therapy	Outcomes Studied	Reported Treatment Effects ^§^
**Ahmed** 2007 New Zealand	Parallel *n* = 80	NR	60–87	*B. lactis HN019* 5 × 10^9^ 1 × 10^9^ or 6.5 × 10^7^	4 w	Changes in fecal microflora	***Bifidobacteria*** ↑ *Lactobacilli* ↑ *Streptococci* ↑ *Coliforms* ↑
**Arunachalam** 2000 New Zealand	Parallel *n* = 25	18/7	60–83	*B. lactis HN019* 1.5 × 10^11^ (twice daily)	6 w	Immunomodulation (and presence of B. lactis in stool)	**IFN-a** ↑ **phagocytic capacity** ↑ bactericidal activity ↑ (both groups)
**Bartosch** 2004 UK	Parallel *n* = 18	18/0	63–90	*B. bifidum BB-02* *B. lactis BL-01* ORAFTI’s Raftilose^®^ Synergy 1 (chicory inulin and oligofructose) ∼3.5 × 10^10^ (of each bacterium)	4 w	Changes in fecal *Bifidobacteria* and Lactobacillus	**Total*****Bifidobacteria*** ↑ ***B. angulatum*** ↑ (***B. bifidum*** ↑) (***B. lactis*** ↑) **Total *****lactobacilli*** ↑
**Costabile** 2017 UK	Crossover *n* = 40	NR	60–80	*L. rhamnosus GG* *pilus-deficient L. rhamnosus GG-PB12* Promitor™ (corn fiber) 12 × 10^10^	3 w (for each supplement)	Changes in fecal microflora, immunity, and blood lipids	***Parabacteroides*** ↑ ***Ruminococcaceae*** ↑ ***Oscillospira*** ↓ ***Desulfovibrio*** ↓ (***L. rhamnosus GG*** ↑) **Total cholesterol** ↓ **LDL-cholesterol** ↓
**Finamore** 2019 Italy	Parallel *n* = 98	29/69	84.6 (mean)	*B. longum Bar33* *L. helveticus Bar13* 1 × 10^9^	30 d	Improvements in innate and adaptive immunity, anthropometrics and wellbeing	**Regulatory T cells** ↑ **B cells **↑ **natural killer activity** ↑ **CD4+ naive T cells** ↑ **CD8+ naive T cells** ↑ **CD8+ activated memory cells** ↑ **CD4+ effector memory cells** ↓
**Gohel** 2016 India	Crossover *n* = 76	38/38	64–74	*L. helveticus MTCC 5463* *S. thermophilus MTCC 5460* (in honey supplemented fermented milk) > 10^8^ CFU/mL *L. helveticus* (dose: 200 mL/day) *S*. *thermo. NR*	4 w	Effect on serum calcium and hematological parameters	**Serum calcium level** ↑
**Guillemard** 2010 France	Parallel (multi-center) *n* = 1072	672 /400	69–95	*L. casei DN-114 001* *S. thermophilus* *L. delbrueckii subsp. bulgaricus* *L. casei* >10^10^ CFU/100 g The other two at >10^9^ CFU/100 g (dose: 200 g/day)	3 m	Resistance to common infectious disease (and prescence of *L. paracasei* in stool)	**Duration of CID** ↓
**Inoue** 2018 Japan	Parallell *n* = 38	24/14	66–78	*B. longum BB536,* *B. breve M-16V* *B. infantis M-63* *B. breve B-3* (dextrin) and moderate resistance training ∼1.25 × 10^10^ (of each bacterium)	12 w	Cognitive function	MOCA-J ↑ (both groups) Defecation frequency↑ Mental state ↓ body mass/BMI ↓ Anxiety ↓ (placebo)
**Kim** 2021 Korea	Parallel (multi-center) *n* = 53	NR	Treatm: 72 Placebo: 71.1 (mean)	*B. bifidum BGN4* *B. longum BORI* 1 × 10^9^	12 w	Intestinal and brain health	**inflammation-causing gut bacteria** ↓ **Mental flexibility** ↑ **BDNF** ↑ (**placebo**)
**Macfarlane** 2013 UK	Crossover *n* = 43	22/21	65–90	*B. longum* ORAFTI’s Raftilose^®^ Synergy 1 (chicory inulin and oligofructose) Ca 2 × 10^11^	4 w	Changes in fecal *Bifidobacteria* counts, changes in fecal microflora, inflammatory markers, bowel habit and health status	***Bifidobacteria*** ↑ ***B. adolescentis*** ↑ ***B. angulatum*** ↑ ***B. bifidum*** ↑ *(**B. longum*** ↑) **Actinobacteria** ↑ **Firmicutes** ↑ **Proteobacteria** ↓ **Butyrate** ↑ Isobutyrate ↑ Acetate ↑ **TNF-a** ↓
**Manzoni** 2017 Brazil	Parallel *n* = 29	NR	Treatm: 67 Placebo: 71 (mean)	*B.**animalis ssp. lactis BB-12* Yacon (prebiotic source) and soy extracts 10^10^ CFU/100 mL (dose: 150 mL/day)	4 w	Changes in counts of fecal: *Bifidobacterium spp., Clostridium spp*., *Enterobacteria*. Additionally, polyamines and inflammatory cytokines	Polyamine levels ↑ (both groups)
**Moro-Gracía** 2013 Spain	Parallel (multi-center) *n* = 47	7/40	Treatm: 65–82 Placebo: 65–90	*L. delbrueckii subsp. bulgaricus 8481* 3 × 10^7^/capsule (dose: 3 capsules/day)	6 mo	Immune cell populations, cytokines, T cell receptor excision circle (TREC), human β-defensin-2 (hBD-2) concentrations, cytomegalovirus (CMV) IgG titers	NK cells ↑ CD8+ T cell ↓ CD4/CD8 ratio ↑ Senescent T cells ↓ NAÏVE CD4+ T cells ↑ Memory T cells ↓ TREC ↑ IL-8 ↓ CMV titers ↑ (placebo)
**Nyangale** 2015 UK	Crossover *n* = 36	25/17	65–80	*Bac. coagulans GBI-30, 6086 (BC30)* 1 × 10^9^	28 d	Immunomodulation, changes in fecal microflora, Calprotectin and SCFA. Additionally, digestive health and mood diaries.	***F. prausnitzii*** ↑ IL-10 ↑ TNF-a ↑ (placebo) SCFA ↑ (both groups)
**Ouwehand** 2009 Finland	Parallel *n* = 47	35/12	Treatm: 70.3 Placebo: 71.7 (mean)	*L. acidophillus NCFM* Lactitol (prebiotic) 2 × 10^9^ CFU/g (dose: 5–5.5 g, twice a day)	2 w	Changes in fecal: *Bifidobacteria* *L acidophilus* *L acidophilus NCFM* *C. perfringes* Sulphate reducers (i.e., *D. intestinalis*)	**Stool frequency** ↑ ***Bifidobacteria*** ↑ **Spermidine **↑ **Fecal PGE**_**2**_ ↑ Fecal IgA ↑ (**L. acidophilus NCFM** ↑)
**Shinkai** 2013 Japan	Parallel *n* = 278	140/138	>65 yrs	*L. pentosus strain b240* 2 × 10^9^ or 2 × 10^10^	20 w	Common cold, QoL	**Common cold incidence** ↓ **General health perception** ↑
**Spaiser** 2015 USA	Crossover *n* = 32	22/10	69.8 (mean)	*L. gasseri KS-13* *B. bifidum G9-1* *B. longum MM2* 1.5 × 10^9/^capsule (Dose: 2 capsules/day)	3 w	Changes in fecal: *Bifidobacteria* Lactic acid bacteria *E. coli* Circulating CD4+ lymphocytes and PHA stimulated cytokine release Digestive health	***Bifidobacteria*** ↑ **Lactic acid bacteria** ↑ **E. coli** ↓ IL-10 ↑ IL-5 ↑ CD4+ ↓ (placebo)
**Östlund-Lagerström** 2016 Sweden	Parallel *n* = 249	152/97	Treatm: 72.6 Placebo: 72 (mean)	*L. reuteri DSM 17938* Rhamnose, galactooligosaccharide and maltodextrin 10^8^ CFU/day	12 w	Digestive health Wellbeing	No significant effects

* *n* Analyzed; # range if nothing else is stated; § arrows imply direction of the treatment effect (↑ = increase, ↓ = decrease). Bold text denotes differences in the treatment group, significant to placebo; otherwise, the arrows refer to differences from baseline in the treatment group if nothing else is stated.

**Table 2 microorganisms-09-01344-t002:** Overview of studies with microbiota composition as an outcome.

Author, Year	Probiotic/Synbiotic	Total study Duration	Sampling Timepoints	Number of Subjects’ Feces Analyzed	Microbiota Assessed	Methods
Synbiotic interventions						
Bartosch [48]	*B. bi**fi**dum BB-02* *B. lactis BL-01* ORAFTI’s Raftilose^®^ Synergy 1 (chicory inulin and oligofructose)	prefeeding (1 week) feeding (4 weeks) postfeeding (week 8)	1 w, 4 w, 8 w	18 *n* = 9 (placebo) *n* = 9 (synbiotic)	*Agar plate cultures*	
					Total anaerobes	Wilkins-Chalgren agar
					*Bifidobacteria*	Beerens medium
					*Lactobacilli*	Rogosa
					*B. lactis, B. bifidum, Bifidobacteria* genus	qPCR (DNA primers)
Costabile [47]	*L. rhamnosus GG* pilus-deficient *L. rhamnosus GG-PB12* Promitor™ (corn fiber)	147 d (2 w run-in, 3 w intervention, 3 weeks washout)	0, 21, 63, 105, 147 d	111 LLG-PB12 + SCF (*n* = 37) LGG + SCF (*n* = 37) SCF (*n* = 37)	Quantities of the *L. rhamnosus GG* strains and total bacteria	qPCR (16s rRNA)
					Phylogenetic analysis	16s rRNA Illumina Miseg sequencing and profiling (V3,V4 variable region sequencing using primers)
Macfarlane [50]	*B. longum* ORAFTI’s Raftilose^®^ Synergy 1 (chicory inulin and oligofructose)	12 w (4 w intervention)	baseline, 2 w (mid-intervention), 4 w (end)	43 (crossover)	Firmicutes (*Clostridium* cluster XIVa, *F. prausnitzii* group, *Ruminococci*, *Roseburia intestinalis*, lactic acid bacteria)	FISH (16s rRNA)
					Bacteroidetes (*Bacteroides*/*Prevotella*), Actinobacteria (Atopobium group, *Bifidobacteria*)	
					Proteobacteria (*Enterobacteriaceae*, *Desulphovibrio*)	
					*Bifidobacteria*	
					Total bacteria (Eubacterial probe)	
Manzoni [59]	*B. animalis ssp. lactis BB-12* Yacon (prebiotic source) and soy extracts	8 w (2 w prefeeding, 4 w feeding, 2 w postfeeding)	1 w, 6 w (end of feeding), 8 w (end of washout)	12 *n* = 6 (intervention) *n* = 6 (placebo)	*Agar plate cultures*	
					Clostridium	Reinforced Clostridial Agar
					Enterobacteriaceae	MacConkey medium
					*Bifidobacteria*	Iodoacetate Medium-25 (BIM25)
Ouwehand [56]	*L acidophillus NCFM* Lactitol (prebiotic)	6 w (2 w run in, 2 w intervention, 2 w washout)	baseline, 2 weeks, 4 weeks	47 *n* = 24 (intervention) *n* = 23 (placebo)	*Bifidobacteria* *L. acidophilus* *L. acidophilus NCFM* *C. perfringes* Sulphate reducers (*D. intestinalis*)	qPCR (16s rRNA probes)
					Total bacteria counts	Flow cytometry
Probiotic interventions						
Ahmed [49]	*B. lactis HN019* (in skim milk)	8 w (2 w run in, 4 w intervention, 2 w washout)	0,2,4,6,7,8 w	66 Bifidobacterium: low (*n* = 18), med (*n* = 15), high (*n* = 19), placebo (*n* = 14)	*Agar plate cultures*	
					*Bifidobacteria*	Beerens medium
					*Lactobacilli*	Rogosa SL Agar
					*Streptococci* *Enterobacteria*	MacConkey agar and bile esulin azide agar
					Total anaerobes	Brucella agar
					Bacteroidetes	Bacteroidetes-bile-esculin
					Yeast and mold	Sabaroud dextrose agar
Arunachalam [52]	*B. lactis HN019* (in skim milk)	6 w	NR	25 *n* = 13 (intervention) *n* = 12 (placebo)	*B. lactis HN019*	RAPD-DNA (strain-specific DNA probe)
					Bacterial cell viability in test product	MRSC Agar plate cultures, *Bifidobacteria*- Beerens’ medium
Guillemard [44]	*L. casei DN-114 001* *S. thermophilus* *L. delbrueckii subsp. bulgaricus* (in fermented dairy product, Actimel)	3 m (84 d)	1,2,3,4 m	63 *n* = 32 (intervention) *n* = 31 (placebo)	*L. paracasei*	qPCR (L. paracasei group specific)
Kim [61]	*B. bifidum BGN4* *B. longum BORI*	12 w	weekly	Not clear	Phylogenetic analysis	16s rRNA Illumina Miseg sequencing and profiling (V3,V4 variable region sequencing using primers)
Nyangale [58]	*Bac. coagulans GBI-30, 6086 (BC30)*	28 d	baseline, 28d	36 (crossover)		FISH (DNA olig probes):
					*Bifidobacterium* spp.	Bif164
					*Lactobacillus* spp. and *Enterococcus* spp.	LAB158
					*Clostridium coccoides*, *Eubacterium rectale* group, (*Clostridium* cluster XIVa and XIVb)	EREC482
					*Clostridium lituseburense* group (*Clostridium* cluster XI)	CLIT135
					*Bacteroidaceae spp., Prevotellaceae spp*., some of *Porphyromonadaceae spp.*	BAC303
					*F. prausnitzii* and relatives	Fprau645
					*Bacillus spp.,* including *B. coagulans*	Bcoa191
Spaiser [54]	*L. gasseri KS-13* *B. bifidum G9-1* *B. longum MM2*	3 w	baseline, 3 w	28 (crossover)	*Bifidobacteria*	qPCR (DNA primers)
					Lactic acid bacteria	
					*E. coli*	
					OTU Classification	pyrosequencing (ESPIRIT tree algorithm for binning for of sequences, similarity levels of 98% and 95%)

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
