# Peer review of "The Effect of Probiotics on Health Outcomes in the Elderly: A Systematic Review of Randomized, Placebo-Controlled Studies"

_microorganisms, 2021, doi:10.3390/microorganisms9061344_

Round 1

Reviewer 1 Report

The manuscript is generally well-written. Some editing needs to be done and some examples are listed below:

  • In the abstract please remove the headings
  • Line 22-24: Please correct to passive voice
  • Line 200: There is an issue with the gap
  • Was the minimum age used in the study the 60 years or 44-60? Please correct if wrong
  • Line 263: Please correct B. Lactis to B. lactis. The same issue throughout the manuscript
  • Line 281: Remove levels
  • Line 320: B. longum
  • Line 335: Please write Lactobacillus with italics
  • Line 339: Full stop missing
  • Line 372: bulgaricus
  • Line 469: r?

Author Response

Thank for your insightful and helpful comments.  We agree with the reviewer that the headings in the abstract are perhaps distracting, and they have now been removed.  The (44-61) actually refers to the reference numbers, not the ages of the participants, but please let us know if this is unclear.  B. Lactis has been changed to B. lactis now as you suggested.  The extra "levels" has been removed.  B longum has bene changed to B. longum.  Lactobacillus has been written with italics.  Bulgaricus has been changed to bulgaricus.

Reviewer 2 Report

The article is interesting, covering a relevant and original topic.

Please, see below some comments:

  1. the introduction is too long and it will benefit from a shortening. Please, try to focus on the main messages. what is really needed for the readers to know in order to be able to understand the content of the manuscript. If you consider all the information provided in the introduction useful, then reconsider moving them in the results or in the discussion. It is not needed to provide so many details in the introduction.
  2. line 153. it is part of the results section. move it in the appropriate section, as well as for figure 1. 
  3. lines 204-207. Please for each of the categories identified add the references.
  4. line 218. The authors stated that included studies were conducted in 12 countries. Can you provide more details? This information is not part of the table1.
  5. table 1. regarding sample size, can you provide information regarding a number of subjects included in the intervention group compared to the control group for each of the included studies? 
  6. Table 1. in the column age, or sample size, can you also specify the sec of included subjects? neither information on which type of intervention is provided. please, add.
  7. lines 225-226 it was already part of the methods. It appears here as a repetition. I suggest removing it from here.
  8. line 559 there is a typo.
  9. several sections of the discussion seem a repetition of the results section. My suggestion, again, is to be more synthetic and to just focus on what was not already said in the results. Be more concise would improve the readability of the work and will help readers to catch your main messages. please revise. 
  10. paragraphs 4.6-4.7 and 4.8 are well written. 

Author Response

Thank you so much for reviewing our manuscript and for your insightful, helpful comments.  We agree with the reviewer's comment that the introduction is too long.  Details have been removed from the third paragraph, and the fourth/fifth paragraph have been shortened are now combined.  Line 153 and Figure 1 are now included in the results.  The reference numbers for the different outcomes examined have now been added.  We agree with the reviewer that more information should be added to Table 1.  Table 1 now includes the country in which the study was conducted (column 1) and sex (column 3).  We desired to include how many participants in each group, but there was an issue with space.  We agree that the first sentence in the assessment of study quality was already in the methods, and we deleted it.  The typo in line 559 has now been corrected.  Thank you for your feedback on the discussion.  We have now significantly revised the discussion, removing redundant detail in order to better highlight the relevant material.